# The intermediate proteasome is constitutively expressed in pancreatic beta cells and upregulated by stimulatory, low concentrations of interleukin 1 β

Muhammad Saad Khilji[1,2], Danielle Verstappen[1,3], Tina Dahlby[1], Michala Cecilie Burstein Prause[4], Celina Pihl[1], Sophie Emilie Bresson[1], Tenna Holgersen Bryde[1], Phillip Alexander Keller Andersen[1], Kristian Klindt[1], Dusan Zivkovic[5], Marie-Pierre Bousquet-Dubouch[5], Björn Tyrberg[6¤], Thomas Mandrup-Poulsen[1], Michal Tomasz Marzec[1] *

1 Laboratory of Immuno-endocrinology, Inflammation, Metabolism and Oxidation Section, Department of Biomedical Sciences, University of Copenhagen, Copenhagen, Denmark, 2 Department of Physiology, University of Veterinary and Animal Sciences, Lahore, Punjab, Pakistan, 3 Radboud Universiteit, Nijmegen, Netherlands, 4 Section for Beta-cell Biology, Department of Biomedical Sciences, University of Copenhagen, Copenhagen, Denmark, 5 Institut de Pharmacologie et de Biologie Structurale, Centre National de la Recherche Scientifique, Université de Toulouse, Toulouse, France, 6 Department of Physiology, Institute of Neuroscience and Physiology, Sahlgrenska Academy, University of Gothenburg, Gothenburg, Sweden

¤ Cardiovascular and Metabolic Diseases, Institute de recherches Servier, Suresnes, France
* Michal@sundk.ku.dk

**Data Availability Statement:** All relevant data are within the manuscript and its Supporting Information files.

## Abstract

A central and still open question regarding the pathogenesis of autoimmune diseases, such as type 1 diabetes, concerns the processes that underlie the generation of MHC-presented autoantigenic epitopes that become targets of autoimmune attack. Proteasomal degradation is a key step in processing of proteins for MHC class I presentation. Different types of proteasomes can be expressed in cells dictating the repertoire of peptides presented by the MHC class I complex. Of particular interest for type 1 diabetes is the proteasomal configuration of pancreatic β cells, as this might facilitate autoantigen presentation by β cells and thereby their T-cell mediated destruction. Here we investigated whether so-called inducible subunits of the proteasome are constitutively expressed in β cells, regulated by inflammatory signals and participate in the formation of active intermediate or immuno-proteasomes. We show that inducible proteasomal subunits are constitutively expressed in human and rodent islets and an insulin-secreting cell-line. Moreover, the β5i subunit is incorporated into active intermediate proteasomes that are bound to 19S or 11S regulatory particles. Finally, inducible subunit expression along with increase in total proteasome activities are further upregulated by low concentrations of IL-1β stimulating proinsulin biosynthesis. These findings suggest that the β cell proteasomal repertoire is more diverse than assumed previously and may be highly responsive to a local inflammatory islet environment.

**Funding:** This study was funded by The Punjab Educational Endowment Fund https://www.peef.org.pk/ (M.S.K), the Department of Biomedical Sciences at the University of Copenhagen https://bmi.ku.dk/ (T.D. and M.T.M.); the Augustinus Foundation https://augustinusfonden.dk/ (T.D. and M.T.M.); EFSD/JDRF/Lilly European Programme in Type 1 Diabetes http://www.europeandiabetes foundation.org/, Vissing Fonden http://www.vissing fonden.dk/, Bjarne Jensen Fonden http://www.bjarnejensensfond.dk/, Poul og Erna Sehested Hansens Fond, no website, Eva og Hans Carl Holms Mindelegat https://haldguttenberg.dk/1-april/(M.T.M). The funders had no role in study design, data collection and analysis, decision to publish, or preparation of the manuscript.

**Competing interests:** The authors have declared that no competing interests exist.

**Abbreviations:** i-proteasome, Immunoproteasome; int-proteasome, Intermediate proteasome; s-proteasome, Standard proteasome; β1, beta subunit 1; β2, beta subunit 2; β5, beta subunit 5; β1i, inducible beta subunit 1; β2i, inducible beta subunit 2; β5i, inducible beta subunit 5; PSMB8, Proteasome subunit beta type-8 = β5i; PSMB9, Proteasome subunit beta type-9 = β1i; PSMB10, Proteasome subunit beta type-10 = β2i; HBSS, Hank's balanced salt solution; IL-1β, Interleukin 1 β; INF, Interferon; MHC, Major histocompatibility complex.

## Introduction

The proteasome is a multi-subunit complex essential for the proteolytic degradation of cellular proteins and in the generation of specific sets of bioactive peptides [1] influencing a variety of cellular processes e.g. transcriptional regulation, signaling and the regulation of the cell cycle progression [2–5].

Proteasomal activity is executed by the proteolytic core, known as the 20S proteasome. It consists of a stack of four heptameric rings: two outer α and two inner β rings [1]. The β rings are composed of catalytically active subunits (β1, β2 and β5) that cleave peptide bonds at the C-terminal side of proteins [6] with caspase-, trypsin- and chymotrypsin-like activities, respectively [7, 8]. The standard 26S proteasome contains a 19S regulatory cap that binds the polyubiquitin chain, denatures the protein, and feeds it into the proteolytic core of the proteasome [9].

Standard proteasomes (s-proteasome) assembled with β1, β2 and β5 subunits are ubiquitously expressed, but specialized proteasomes also exist and are constitutively expressed by e.g. immune cells [9], where they represent the dominant form. Formation of the proteolytic core of these specialized proteasomes involves substitution of the constitutively expressed catalytic β1, β2 and β5 subunits with the interferon (IFN)-γ-inducible β1i, β2i and β5i subunits (alternatively termed Psmb9/LMP2, Psmb10/MECL-1/LMP10 and Psmb8/LMP7, respectively) [6, 10, 11]. The immune-proteasome (i-proteasome) has an alternative 20S catalytic core where all β-subunits are replaced by IFN-γ inducible β-subunits and where the 20S-associated 19S can be replaced by the 11S (also termed PA28αβ) proteasome regulator [9, 12, 13].

When standard and inducible subunits are present in cells, the latter are preferentially incorporated into newly produced 20S proteasomes [14, 15]. Interestingly, co-expression of standard and inducible β subunits enables cells to assemble a variety of distinct 20S complexes, collectively referred to as intermediate proteasomes (int-proteasomes) [9]. The two most common int-proteasomes are composed of two inner rings containing either β1/β2/β5i or β1i/β2/β5i. These int-proteasomes are not exclusive, as other combinations have been observed, including 20S proteasome with one constitutive (β1/ β2/ β5) and one immune (β1i/ β2i/ β5i) inner ring (also called asymmetric proteasomes, [16–18]).

Immune cells permanently and many other cells under conditions of oxidative stress, inflammation, cytokine stimulation, or viral and bacterial infection express and assemble i- and int-proteasomes [9, 19]. Recently, induction of expression of such proteasomes upon exposure of human pancreatic islets and rat and mouse insulinoma cells to INFγ and β but not high concentrations of IL-1β, was reported [18, 20]. Furthermore, int-proteasomes (but not i-proteasomes) are constitutively expressed in various cells, including liver, heart, kidney, lung or colon [16, 21–24]. They constitute between 1% (heart) to 50% (liver) of the total proteasome pool [16, 21, 23, 24].

The proteasomal composition in cells has broad implications, as proteasomes exhibit diverse substrate specificities. This affects the peptide repertoire generated for presentation on major histocompatibility complex (MHC) class I molecules [13, 19, 25], signal transduction via e.g nuclear factor kappa-light-chain-enhancer of activated B cells (NFκB) [26] and protein degradation e.g. of proinsulin [27].

The s-proteasome is known to improve glucose-stimulated insulin secretion [28], regulate intracellular proinsulin levels [27] or protect against lipotoxic endoplasmic reticulum stress [29]. However, the functions of i- and int-proteasomes are poorly defined. Importantly, *constitutive* expression of inducible proteasome subunits in pancreatic β cells has not been described, but their induction upon INFγ and β treatment has been suggested to play a protective role against cytokine-induced apoptosis [20] and during antiviral responses [18].

Of special interest to type 1 and 2 diabetes pathogenesis is the constitutive profile of the β cell proteasomes and their regulation. Type 1 diabetes (T1D) is an autoimmune disease, in which tolerance to β cells is broken, with proinsulin serving as a major autoantigen. T1D is histologically characterized by pancreatic islet inflammation with increased levels of cytokines i.e. IL-1β, INF-γ/β and TNF-α, in the islet microenvironment [30]. Type 2 diabetes (T2D) arises when insulin secretion fails to meet demands mainly due to impaired insulin sensitivity, with β-cell oxidative and endoplasmic reticulum stress, lipotoxicity and glucotoxicity as consequences causing progressive loss of β cell functional mass [31]. All these cellular stresses induce an inflammatory response or are exacerbated by or associated with low-grade systemic inflammation via production of interleukin 1β (IL-1β) and IL-6 and recruitment and activation of innate immune cells [32, 33]. As i- and int-proteasomes can modify e.g. signal transduction and MHC I peptide presentation, their constitutive and/or induced expression in β cells by inflammatory cytokines is of high interest and therapeutic potential.

Here, we hypothesized that β cells constitutively express active non-standard proteasomes and that the expression is upregulated by innate inflammatory signals at low levels. We therefore set out to analyze the composition of proteasomes in human and mouse islets as well as in the commonly used β-cell model INS-1E cell line in non-stimulated or cytokine-stimulated conditions. We report constitutive transcription and translation of inducible proteasome subunits (β1i/ β2i/ β5i) in β-cells, albeit with lower expression levels compared to immune cell-lines. Of the inducible subunits, β5i is incorporated into active proteasomes in non-stimulated INS-1E cells, forming intermediate proteasomes that constitute 14% of total proteasomes in these cells. Furthermore, mRNA and protein expression of inducible subunits is upregulated by *low* concentrations of IL-1β. β5i and β1i subunits were induced in all tested cellular models while β2i was induced in mouse (but not human) islets and INS-1E cells. Consequently the composition and both constitutive and stimulated activity of proteasomes in β cells has to be considered when investigating degradation mechanisms and antigen presentation on MHC I molecules of proinsulin and other β-cell proteins.

## Materials and methods

### Cell culture

The rat insulinoma INS-1E cell line, a gift from Claes Wollheim and Pierre Maechler, University Medical Center, Geneva, Switzerland, was maintained as previously described [11]. The mouse insulinoma MIN6 cell line, was cultured in DMEM (Life Technologies, Nærum, Denmark) with 25 mM glucose, supplemented with 10% FBS, 0.1% Penicillin/Streptomycin (P/S), 50 uM β-mercaptoethanol and 2 mM L-glutamine. The mouse lymphocyte cell line A20, donated by Prof. Søren Buus, Department of Immunology and Microbiology, University of Copenhagen, Denmark, was cultured in RMPI-1640 (Life Technologies, Nærum, Denmark), containing 10% FBS, 1% P/S, 10 mM HEPES, 50 uM β-mercaptoethanol and 4.5 g/L D-glucose. The human T lymphocyte cell line Jurkat, also from Prof. Buus, was cultured in RPMI-1640 with 10% FBS and 1% P/S. All cells were maintained at $37^{o}$ C with 5% $CO_2$. All cell-lines were *Mycoplasma* negative.

### Animal Care

B6 C57BL/6NRJ mice were housed, handled and sacrificed according to Danish legislation for animal experimentation and with prior approval from the local animal ethics committee, issued by the Department of Experimental Medicine, University of Copenhagen. Animal handling and procedures were conducted by researchers with FELASA certification and supervised by veterinarians.

**Table 1. Human islets donors information and islet preparations used during the investigation.**

| Islet donors information | | | | |
|---|---|---|---|---|
| Donor | 1 | 2 | 3 | 4 |
| Age | 63 | 62 | 58 | 20 |
| Gender (M/F) | F | F | M | M |
| BMI | 19.5 | 29.3 | 27.8 | 21.8 |
| blood group | A+ | A+ | O+ | B+ |
| HLA (A:B) | 2,11 : 18,57 | 26,29 : 7,18 | 2,26 : 35,55 | 11,24 : 18,51 |
| HLA (DR) | 11,17 | 4,15 | 14,16 | 1,11 |
| Cold ischemia time (h) | 9 | 5 | 8.5 | 8 |
| Islets culture duration (h)* | 16 | 14 | 20 | 42 |
| Cause of death | Cerebral bleeding | Cerebral bleeding | Cerebral bleeding | Anoxia |
| Source of islets | ECIT | ECIT | ECIT | ECIT |
| Estimated viability (%) | 95 | 95 | 95 | 95 |
| Estimated purity (%) | 90 | 90 | 90 | 90 |
| Any additional note | EBV positive | | | EBV positive |

*time from islets isolation to shipment

## Islet isolation and culture

Mouse islets were isolated by injection of LiberaseTM TL (Roche®, Hvidovre, Denmark) through the common bile duct to digest exocrine tissue. Islets were handpicked and either lysed immediately or cultured for 3–5 days in RPMI-1640 supplemented with 10% FBS and 1% P/S, at 37° C and 5% $CO_2$. All data points represent separate islet collections (tested in technical triplicates) and thus denote biological variability.

Human islets were isolated from healthy, heart-beating donors by the European Consortium for Islet Transplantation (ECIT) in Milan, Italy, with local ethical approval. The obtained islets were ~ 90% pure and no apparent difference in their quality was observed. Details on islet donors are included in the Table 1. Islets were cultured as previously described in [11].

## Cytokine exposure

INS-1E cells were exposed to 10 ng/mL rat IFN-γ (R&D, Minnesota, USA) or 15 or150 pg/mL rat IL-1β (BD Bioscience, Lyngby, Denmark) or control medium for 24h. Human islets were exposed to 10 ng/ml human IFN-γ (BD Bioscience, New Jersey, USA) or 30 or 300 pg/ml rat IL-1β, while mouse islets were exposed to either 10 ng/mL rat IFN-γ or 50 or 300 pg/mL rat IL-1β or control medium, both for 24 hours prior to experiments.

## Western blotting

Prior to experiments cells or islets were lysed in lysis buffer, consisting of 100 mM Tris (pH 8.0), 30 mM NaCl, 10 mM KCl, 10 mM $MgCl_2$, 2% NP-40, 20 mM iodoacetamide and protease inhibitor cocktail (Life Technologies, Nærum, Denmark). Protein concentrations were measured using Bio-Rad Protein Assay Dye Reagent (Bio-Rad, Copenhagen, Denmark). Indicated amounts of proteins were loaded on Nu-Page 4–12% bis-tris gels (Thermo Fisher Scientific, Hvidovre, Denmark), and proteins were separated by SDS-PAGE. Gels were transferred to PVDF membranes using the iBLOT2 system (Thermo Fisher Scientific, Hvidovre, Denmark). Membranes were cut prior to incubation with primary antibodies (Table 2) overnight. Primary antibodies were diluted in 2% BSA in TBST (50 mM Tris pH 8, 150 mM NaCl, 0.1% Tween).

**Table 2. Primary and secondary antibodies used during the investigation.**

| Antibody target | Company | Cat# | Dilution |
|---|---|---|---|
| β1i | Abcam | ab243556 | 1:10.000 |
| β5i | Abcam | ab3329 | 1:5.000 |
| β2i | Abcam | ab183506 | 1:1.000 |
| Tubulin | Sigma | T6074 | 1:10.000 |
| Actin | Thermofisher Scientific | MA5-11869 | 1:15.000 |
| Insulin | Cell signaling | 8138S | 1:5.000 |
| Anti-mouse secondary | Cell signaling | 7076S | 1:10.000 |
| Anti-rabbit secondary | Cell signaling | 7074S | 1:10.000 |

Membranes were blotted with appropriate secondary antibodies for 1 hour. Blots were developed using chemiluminescence and captured using the Azure®Saphire Biomolecular Imager. Western blots were quantified using ImageJ software (v. 1.52a, [34]).

## Proteasome activity

INS-1E, A-20 and Jurkat cells, human and mouse islets were plated in duplicates or triplicates in 96-well plates and treated with 50 nM ONX-0914, a selective inhibitor of the β5i subunit activity (Selleck Chemicals, Rungsted, Denmark, IC50: ~10 nM for β5i, [35]) or 2 μM MG132, a broad proteasome inhibitor (Sigma-Aldrich, Søborg, Denmark) or control medium for 2 hours prior to experiments. Chymotrypsin-, trypsin- and caspase-like activity was measured through luminescent assay using commercially available Proteasome-Glo™ Assay (Promega, Nacka, Sweden) according to the manufacturer's protocol. Depicted data are averages of either technical duplicates or triplicates as indicated. The added trypsin-like, chymotrypsin-like and caspase-like activity is referred to as total proteasome activity.

## Bulk mouse islet RNA Sequencing

Five hundred mouse islets were plated and exposed to IL-1β (50 pg/mL) for 10 days or left non-exposed for 10 days. Total mRNA was extracted from the islets by employing RNeasy ® Micro Kit (Qiagen, Vedbæk, Denmark). Single-stranded, single-end sequencing libraries were generated using 35 ng of extracted RNA by means of TruSeq® Stranded mRNA Library Prep (Illumina®, Copenhagen, Denmark), and library sequencing was done with the HiSeq 4000 System (Illumina®, Copenhagen, Denmark). Sequence files were drawn to the UCSC mouse genome NCB137/mm9. Further technical and analysis details in [36] and RNA-seq raw data are accessible here: https://www.ncbi.nlm.nih.gov/geo/query/acc.cgi?acc=GSE110691. In brief, expression levels of all genes were estimated by Cufflink (cufflinks v2.2.1,-p 6 -G $gtf_file—max-bundle-frags 1000000000) using only the reads with exact matches. Since specific mRNA levels were analyzed no correction for multiple testing was done. Results (RPKM) for the specific genes of 3 independent experiments were analyzed by Student's paired t-test, n = 3. RPKM for each gene is provided in Table 3.

## Single-Cell RNA Sequencing of Pancreatic Islets

Each single-cell transcriptome was sequenced to ∼750,000 reads, sufficient for cell-type classification. Islet cell subpopulations were analyzed for *PSMB8*, *PSMB9* and *PSMB10* genes expression using published human islet single-cell sequencing data [37]. FastQ files were downloaded from ArrayExpress (accession: E-MTAB-5061). Data was analyzed with bcbio-nextgen (https://github.com/chapmanb/bcbio-nextgen), using the hisat2 algorithm [38] to

**Table 3. Low concentrations of IL-1β induce β subunit mRNA expression in mouse islets.**

| Gene | Ctrl 1 | Ctrl 2 | Ctrl 3 | IL-1B 1 | IL-1B 3 | IL-1B 3 | Ctrl mean | IL-1β treatment mean | P value |
|------|--------|--------|--------|---------|---------|---------|-----------|----------------------|---------|
| β5i (PSMB8) | 9.38 | 8.54 | 9.03 | 28.13 | 24.57 | 26 | 8.98 | 26.23 | 0.0021 |
| β1i (PSMB9) | 6.70 | 5.41 | 6.59 | 13.31 | 16.02 | 15.79 | 6.23 | 15.04 | 0.0172 |
| β2i (PSMB10) | 12 | 11.75 | 9.57 | 33.99 | 36.21 | 32.18 | 11.11 | 34.13 | 0.001 |
| β5 (PSMB5) | 34.53 | 37 | 38.27 | 42.61 | 42.78 | 34.31 | 36.6 | 39.9 | 0.4657 |
| β1 (PSMB6) | 102.84 | 93.47 | 90.64 | 93.09 | 109.07 | 98.63 | 95.65 | 100.26 | 0.6015 |
| β2 (PSMB7) | 79.63 | 76.68 | 79.96 | 85.41 | 83.28 | 72.34 | 78.76 | 80.34 | 0.7634 |

Upregulation of inducible proteasome subunits upon prolonged, low-dose exposure to IL1-β. Five hundred mouse islets were cultured and exposed to IL-1β (50 pg/mL) for 10 days. Total mRNA was extracted, and bulk (whole pancreatic islets) sequenced and genes identified using the UCSC mouse genome NCB137/mm9. mRNA levels of *PSMB8*, *PSMB9* and *PSMB10* for inducible subunits β5i, β1i and β2i, respectively, were significantly (p = <0.0005 each) upregulated by IL-1β exposure while *PSMB5*, *PSMB6* and *PSMB7* for corresponding standard proteasome subunits β5, β1 and β2 remained unchanged. Data presented as RPMK for individual islet collection (biological replicates) in the respective conditions and their means. Results were analyzed by Student's paired t-test, n = 3.

align sequence reads to human genome version hg38 and uniquely aligned reads within RefSeq gene annotations were used to quantify gene expression with the Salmon algorithm [39]. Data is then expressed as log2 of counts per million (CPM). Only cells that passed the quality control in the original study [37] were maintained for further analysis, and the cell type classification from the original study was also maintained.

## Mass spectrometry for proteasome composition analysis

INS-1E cells were grown to 90% confluence in T175 flasks. The cells were washed with HBSS before incubation with pre-warmed culture media complemented with 0.1% formaldehyde for cross-linking for 15 minutes. Next, 125 mM glycine was added for 10 minutes at 37˚ C to quench the formaldehyde. The advantages of live cell cross-linking vs non-crosslinking step has been evaluated in [40]. The cells were then washed three times with HBSS and centrifuged, and pellets were stored at -80˚C for later proteasome composition analysis. Immuno-purification of the proteasomes from the *in-vivo* cross-linked lysates, was performed as previously described [41]. Briefly, proteasomes were purified by incubating the lysates with CNBr sepharose beads (GE Healthcare) covalently bound to the antibody specific for the α2 subunit of the proteasome (MCP21) (100 mg of beads for 0.8 mg antibody), using 150 million cells per 50 mg of grafted beads. The supernatant was collected, and the beads were washed three times with 40 bead volumes of washing buffer (20 mM Tris-HCl pH 7.6, 1 mM EDTA, 10% glycerol, 150 mM NaCl, 0.1% NP-40, 2 mM ATP and 5 mM MgCl2). Finally, proteins were eluted with 0.5 ml of elution buffer (20 mM Tris-HCl pH 7.6, 1 mM EDTA, 10% glycerol, 3 M NaCl, 2 mM ATP and 5 mM $MgCl_2$). Two additional cycles of purification were conducted, reincubating the collected supernatant with antibody-grafted beads. All fractions were pooled. LC-MS/MS analysis was performed as previously described [21, 42]. Briefly, immuno-purified proteasome samples were precipitated with 20% trichloroacetic acid (TCA), washed with cold acetone and then denatured by boiling at 95˚C for 30 min in the Laemmli buffer, also reversing the cross-links [41]. Proteins were alkylated and concentrated on 12% acrylamide SDS-PAGE gel as a single band, which was cut and washed. Trypsin digestion was then performed overnight at 37˚C and the peptides were extracted from the gel. The digestion mixture was then dried in a Speed-Vac and resuspended with 2% acetonitrile, 0.05% trifluoroacetic acid. The peptide

mixture was then analyzed by nano-LC-MS/MS using an UltiMate3000 system (Dionex) coupled to Orbitrap Fusion mass spectrometer (Thermo Fisher Scientific, Bremen, Germany). Proteins identification, validation and relative quantification were performed as previously published [41].

### Statistical analysis

All samples were selected without bias and represent biological not technical variations. Distribution of islets, specifically, were randomized and independent of e.g. size and shape. As a result, samples should be homogenous and represent biological variation, and both protein expression and activity is therefore assumed to be normally distributed [43, 44]. Furthermore, normality of all expression data was tested with a Shapiro-Wilk test and found normally distributed and tested using a student t-test. Meanwhile proteasome activity and cell viability each data point is represented by a mean value of technical replicates, and as such should be normally distributed according to the central limits theorem [45]. Differences between two groups were assessed by a two-tailed Student's t-test. All statistical analyses were done using GraphPad Prism (v. 6, La Jolla, CA). Data is represented as means ± SD or SEM. P-values of ≤0.05 were considered significant.

## Results

### Inducible proteasome subunits are constitutively expressed in pancreatic islets and β-cell lines

To investigate whether proteasome inducible β subunits are expressed in non-stimulated β cells we re-analyzed RNA-sequencing data of single-cells dispersed from pancreatic islets from healthy individuals [37] and found that between 3.5 to 40% of β, α and δ cells express constitutively mRNA of all inducible subunits (Fig 1A and 1B).

Next, we lysed human and mouse islets, INS-1E (β-cell insulinoma), A20 (B cell lymphoma) and Jurkat (T cell leukemia) cells and analyzed their protein contents by SDS-PAGE and Western blotting. As expected, immune cell lines (A20 and Jurkat) showed high expression of the inducible subunits (Fig 1C–1E). Interestingly, we detected relatively low but consistent expression of all three inducible subunits (β1i, β2i and β5i) in human and mouse islets and INS-1E cells (Fig 1C–1E).

### INS-1E cells contain two major types of proteasomes

We next investigated the composition of proteasomes in INS-1E cells through immunoprecipitation (IP) of the 20S α2 proteasome subunit that is an obligatory member of all types of assembled proteasomes [46].

*In vivo* gently cross-linked INS-1E cells proteasomes were immunoprecipitated and samples analyzed by liquid chromatography coupled to tandem mass spectrometry (LC-MS/MS). About 86% of the total proteasomes were found to be s-proteasomes with enzymatically active rings composed of β1-β2-β5 subunits (Fig 2). However, almost 14% of the precipitated proteasomes contained β5i replacing the standard β5 subunit, forming an intermediate proteasome β1-β2-β5i. We found neither β1i nor β2i in protein complexes precipitated with α2 subunit.

Finally, our MS data indicated the presence of proteasomal regulatory particles 19S and 11S (PA28αγ) within α2 proteasome complexes at the rate of 55.8% and 5.1%, respectively (Fig 2). However, with our experimental approach we cannot assign proteasome types to the detected specific regulatory particles. The remaining 39.1% of 20S corresponds to free (unactivated) proteasome [21, 41].

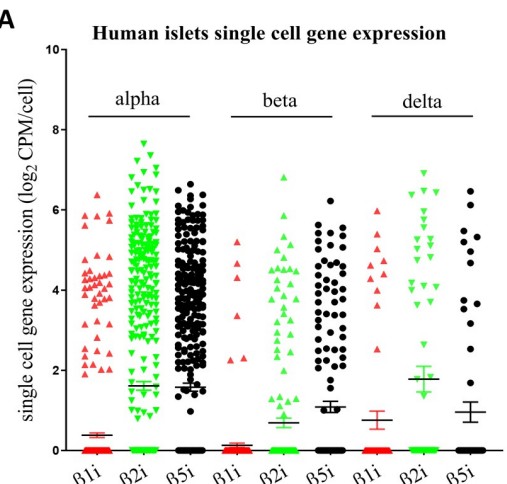

**B**

| Cell type | % of cells positive for inducible subunits | | |
|---|---|---|---|
| | β5i | β1i | β2i |
| Alpha | 39.96 | 9.7 | 37.25 |
| Beta | 29.83 | 3.5 | 19.88 |
| Delta | 22.03 | 16.95 | 38.98 |

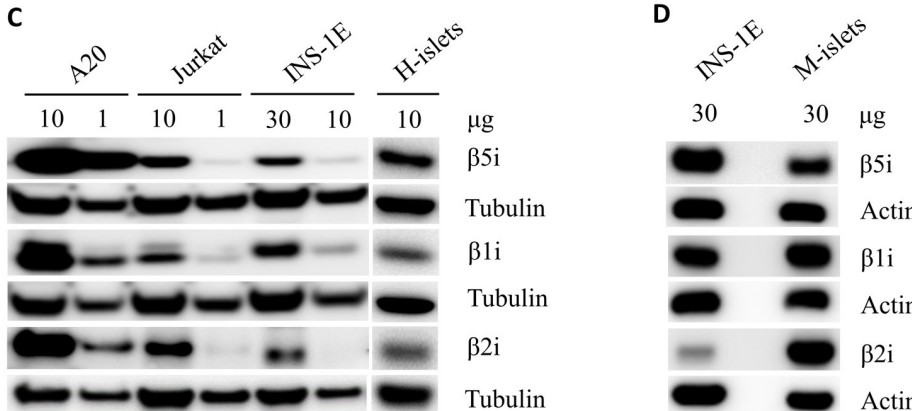

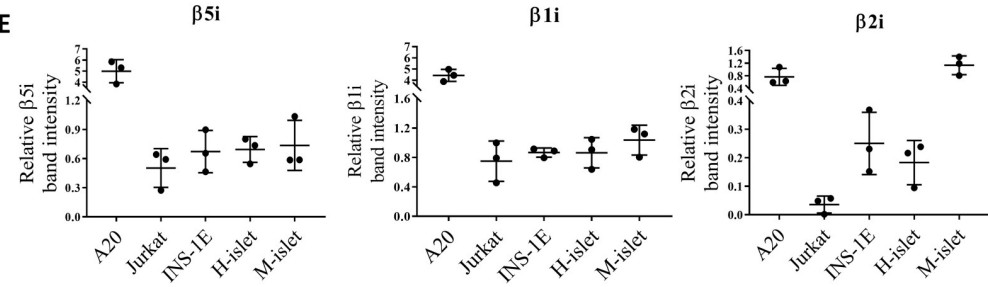

**Fig 1. Constitutive expression of proteasome inducible subunits in islets and cell lines. (A-B)** Single cell RNA sequencing analysis of β1i, β2i and β5i gene expression in human pancreatic islet alpha, beta and delta cells from healthy individuals (n = 6). The data is shown as means with SEM. **(B)** presents the percentage of cells with detectable levels of inducible subunit mRNA. **(C-D)** SDS-PAGE and Western blot analysis of basal expression of proteasome inducible subunits in immune cell lines A20 and Jurkat, in insulinoma beta cell line INS-1E, and human islets (H-islets) and mouse islets (M-islets). Values on top of the Western blots show the amount of protein loaded. C and D are representative blots of n = 3. **(E)** Quantification of relative expression levels of inducible proteasome subunits normalized to tubulin (C) or actin (D) in tested cell lines/islets (n = 3, biological replicates). The data is shown as means with SD.

Mass spectrometry analysis of active proteasomes in INS-1E cells

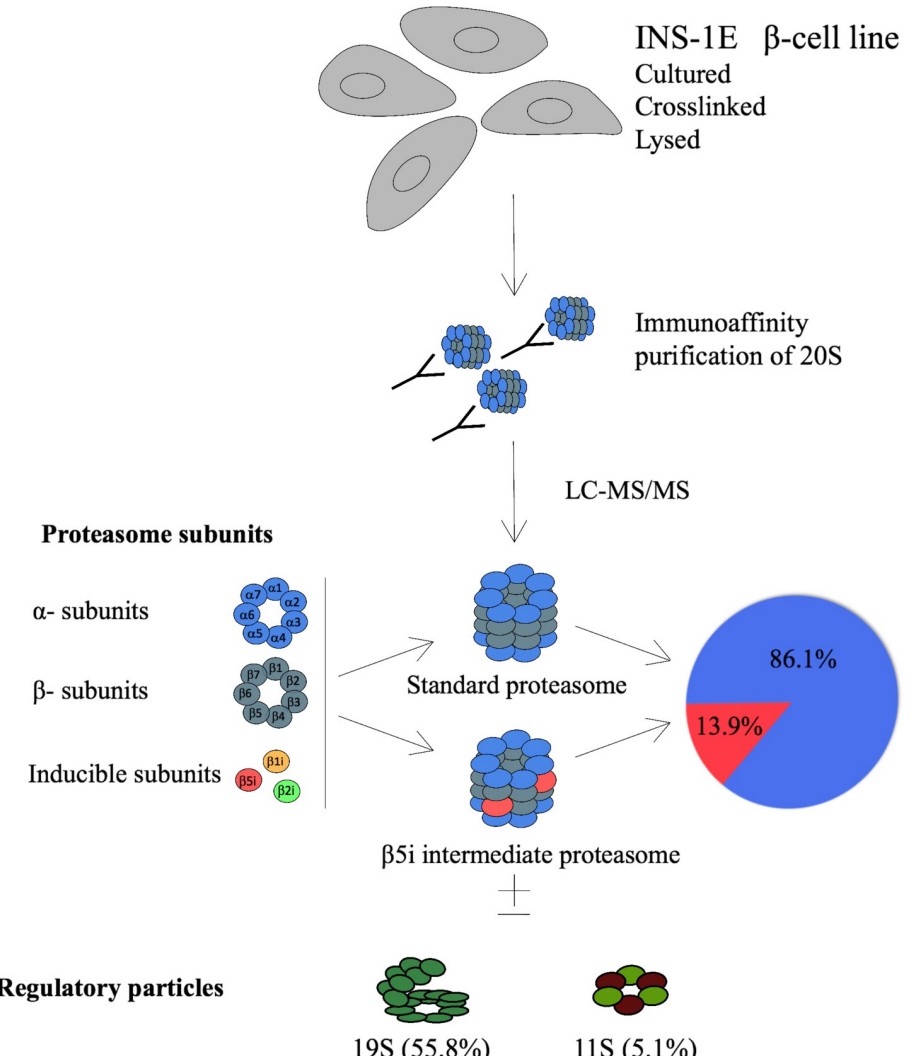

**Fig 2. Identification of intermediate proteasomes in INS-1E cell line.** Four x10^8 cells were cross-linked, lysed, their proteasomes immunoprecipitated with mAb MCP21 and analyzed by LC-MS/MS. The absolute quantities of each of the six catalytic subunits measured by the LC-MS/MS method were computed to calculate the stoichiometry of 20S proteasome subtypes and the fractions of regulatory particles associated with the 20S core particle, as detailed in Experimental Procedures. INS-1E cells were cultured at standard conditions and four biological replicates were analyzed.

## The β5i-selective small-molecule inhibitor ONX-0914 reduces chymotrypsin-like activity of the β cell proteasome

We next examined the profile of proteasome proteolytic activities: chymotrypsin-, trypsin- and caspase-like. The proteolytic activities were tested in unstimulated live cells by addition of specific substrates to the medium (Cell-Based Proteasome-Glo™ Assay). As shown in Fig 3A–

Basal proteasome activity profile

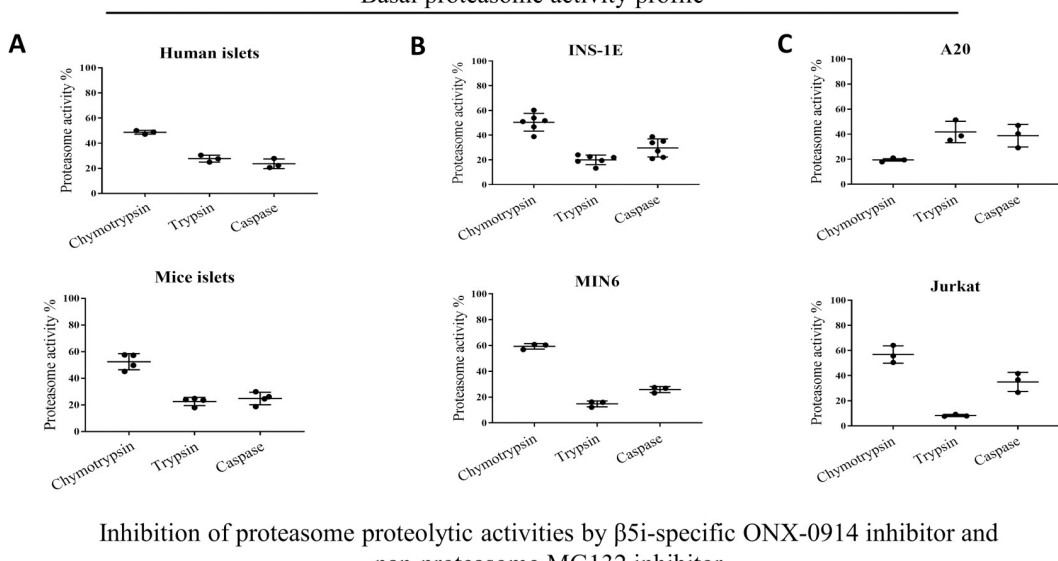

Inhibition of proteasome proteolytic activities by β5i-specific ONX-0914 inhibitor and pan-proteasome MG132 inhibitor

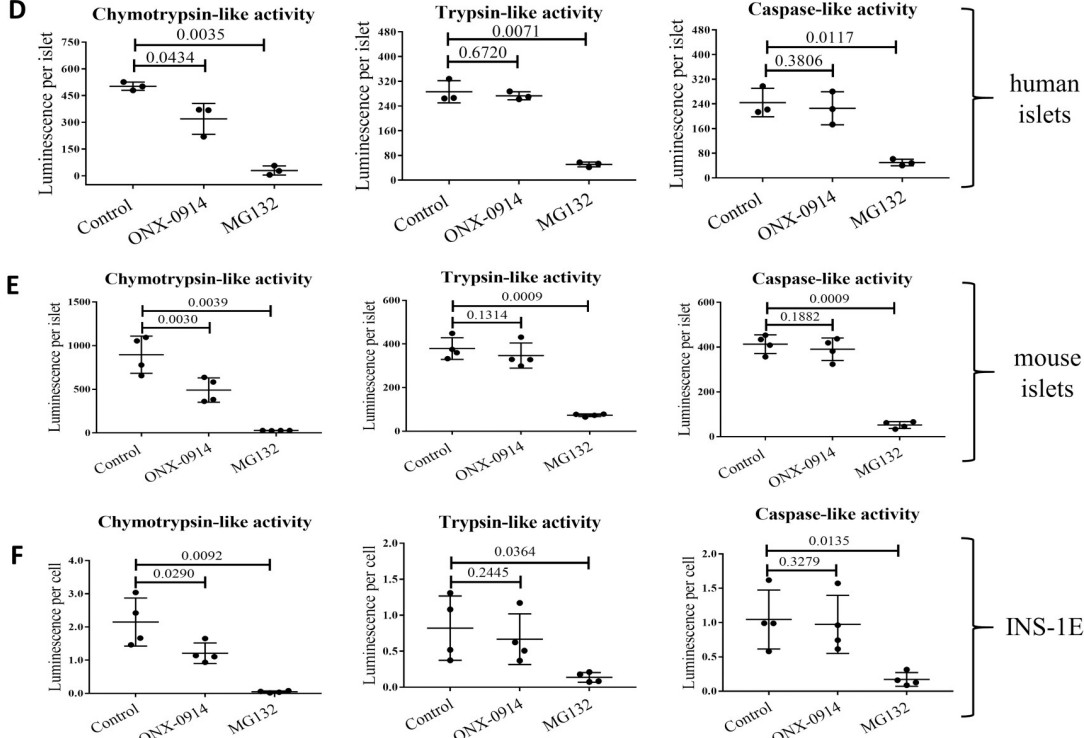

**Fig 3. Total and proteolytic-specific enzymatic activities of proteasomes.** (A) Proteasome activity in human (n = 3) and mouse islets (n = 4), (B) beta cell lines: INS-1E (n = 6) and MIN6 (n = 3) and (C) immune cells: A20 and Jurkat (n = 3). All presented data points are represent biological replicates. Proteolytic-specific activities exhibited by proteasomes subunits, treated with β5i subunit specific inhibitor ONX-0914 (2 h, 50 nM) or non-specific proteasome inhibitor MG132 (2 h, 2 μM), in (D) human islets (n = 3), (E) mouse islets (n = 4) and (F) INS-1E cells (n = 4). Proteasome activity was evaluated in cultured cells/islets using Promega Proteasome Glo assay. The data is shown as luminescence per islet/cell. Statistical analysis was performed by paired t-test of treatments versus control. The data is presented as means with SD.

3C intact human and mouse islets, INS-1E, MIN6 and Jurkat cells exhibited strikingly similar profiles. Under basal conditions, chymotrypsin-like activity constituted 50–60% of total proteasome activity, with the remaining 40–50% of activity almost equally divided between trypsin- and caspase-like activities. In A20 cells, trypsin- and caspase-like activities constituted a larger part of the proteasome activity than did chymotrypsin-like activity, despite the fact that they expressed the highest amounts of β5i and β1i (both subunits have chymotrypsin-like activities).

We next probed what portion of the observed proteasome chymotrypsin-like activity can be attributed to the inducible subunit β5i. We took an advantage of a selective and potent β5i subunit inhibitor, ONX-0914. Pretreatment of human and mouse islets and INS-1E cells with 50 nM ONX-0914 for 4 hours reduced chymotrypsin-like activity by 40%, indicating that β5i is enzymatically active in those islets and cells (Fig 3D–3F). At the same time, trypsin- and caspase-like activities were not affected by ONX-0914. As expected two μM of MG-132 treatment (broad proteasome inhibitor) almost completely blocked all three types of enzymatic activities in all tested cells and islets.

## Low concentrations of IL-1β upregulate β1i, β2i and β5i subunit expression in β cells

Previous work has shown that the β1i, β2i and β5i proteasome subunits are expressed in response to IFN-γ/β in cells other than that of hematopoietic origin including β-cells [20, 21, 41] but their expression was not regulated by the high concentrations of IL-1β treatment [20]. Here we asked, if a similar to IFN-γ/β expression upregulation can be achieved by mimicking low-grade inflammation with the application of a low stimulatory concentration of IL-1β [36].

Mouse islets exposed to 50 pg/ml of IL-1β for 10 days exhibited a significantly higher mRNA expression of β5i (*Psmb8*), β2i (*Psmb9*) and β1i (*Psmb10*) compared to the untreated islets (Table 3). The mRNA levels for β5 (*Psmb5*), *β2 (Psmb6) and β1 (Psmb7)* genes that encode standard subunits remained unchanged after the same exposure.

Furthermore, exposure of human and mouse islets, as well as INS-1E cells, to low concentration of IL-1β for 24 h (15 pg/ml for INS-1E, 30 pg/ml for human islets and 50 pg/ml for mouse islets) induced expression of β1i, β2i and β5i (Fig 4A–4C) with the exception of β2i in human islets. The high concentration of IL-1β (150 pg/ml for INS-1E and 300 pg/ml for human/mouse islets) further induced β1i and β5i expression (but not β2i) in INS-1E cells, but failed to induce upregulation in the subunit expression in human and mouse islets (Fig 4A and 4B). As expected, low IFN-γ treatment for 24 h induced expression of all inducible subunits (Fig 4A–4C). Concentrations of IL-1β in the low range are known to increase insulin biosynthesis [47, 48]. Interestingly, induction of inducible proteasome expression by low concentrations of IL-1β or IFN-γ was associated with increased proinsulin expression levels in INS-1E cells while high concentrations of IL-1β diminished proinsulin expression (S1A and S1BFig). Furthermore, low concentration of IL-1β did not decrease the viability of mouse islets and INS1-E cells over the 24 h exposure to the cytokine (S1C and S1D Fig).

## Low concentrations of IL-1β increase proteasome activity in β cells

INS-1E cells exposed to either low concentrations of IL-1β or IFN-γ showed significant increase in all three proteasome catalytic activities. Furthermore, high concentration of IL-1β increased chymotrypsin-like and trypsin-like activity compared to controls, although the increase was less significant than that observed for low concentration exposure (Fig 5A). Mouse and human islets showed a similar pattern of increasing chymotrypsin-, trypsin- and caspase-like activity, when exposed to low concentrations of IL-1β or IFN-γ (Fig 5B and 5C).

Cytokine induced upregulation of inducible proteasome subunits in islets/cells

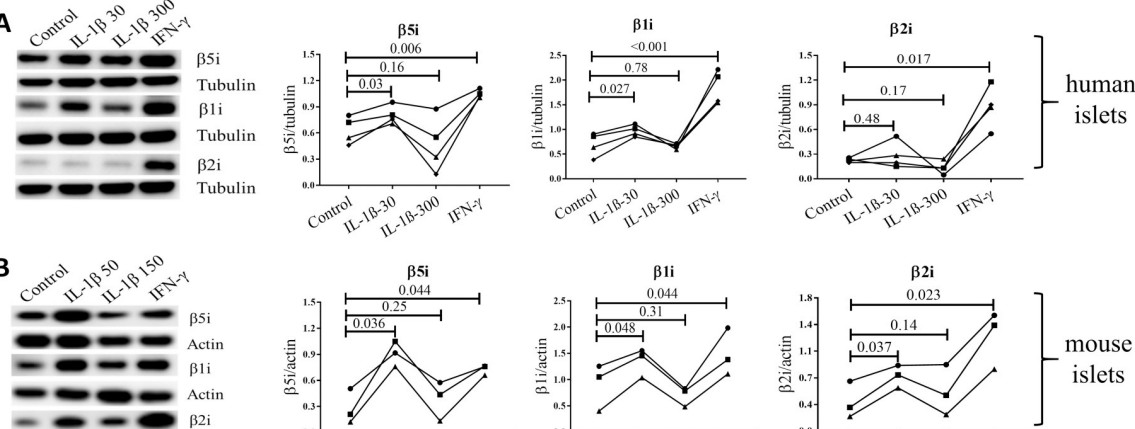

**Fig 4. Cytokines induce upregulation of inducible proteasome subunits in islets/cells.** Human (**A,** n = 4) and mouse islets (**B,** n = 3) and INS-1E cells (**C,** n = 4) were exposed for 24 h to IL-1β at low (50 pg/ml for mouse islets and 15 pg/ml for INS-1E) or high dose (300 pg/ml for human/mouse islets and 150 pg/ml for INS-1E) or IFN-γ (10 ng/ml). Islets/cells were lysed and protein content analyzed by SDS-PAGE and Western blotting. Representative blots of four independent experiments (biological replicates) are shown (left) and quantification of inducible subunit bands relative to the tubulin (A and C) or actin (B) is presented (right). Statistical analysis was performed by paired t-tests of treatments versus control. Experiments done on individual islet donors (A and B) or biological cell replicates (C) are connected by lines.

A high concentration of IL-1β did not have a significant effect on any of the proteasome-based catalytic activities in mouse and human islets.

## Discussion

The present work shows that 1) inducible proteasome subunits are constitutively expressed in human and rodent islets and a β-cell line, 2) β5i is incorporated into an active proteasome, forming int-proteasomes and 3) inducible subunit expression is upregulated by low IL-1β concentrations. The cellular composition of proteasomes and their expressional regulation is of particular interest, because different types of proteasomes degrade proteins and peptides with different efficiency and specificity [49] influencing a variety of cellular processes including antigen presentation and thereby maintenance of peripheral tolerance or induction of autoimmunity [8, 13].

The presence of int-proteasomes as normal constituents in different tissues has been established before as they have been reported to constitute up to 50% of the total proteasome pool, depending on the tissue [16, 24], but a comprehensive investigation of proteasome composition in primary β cells or β cell models in non-stimulated conditions has not been performed.

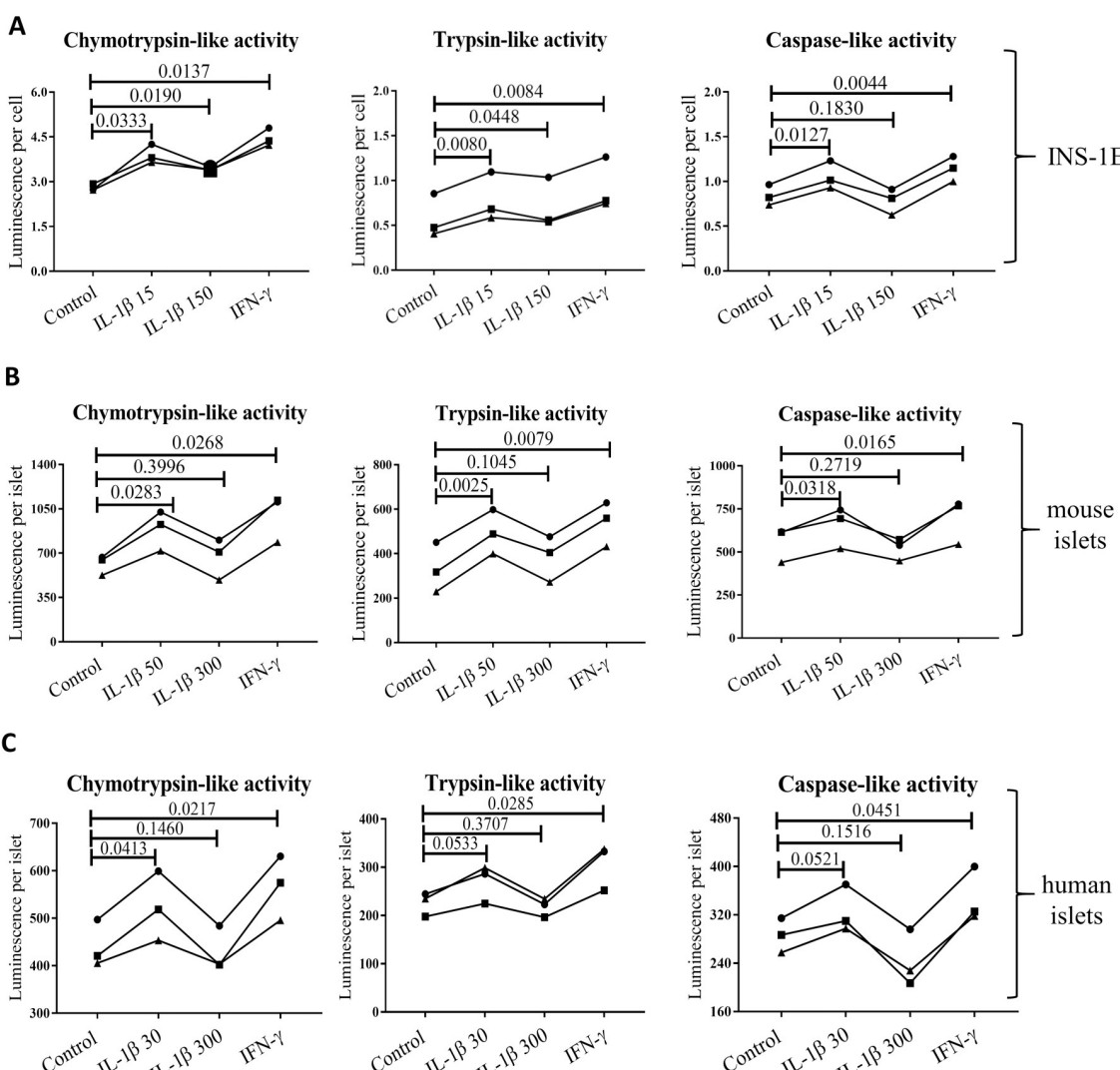

**Fig 5. Basal and cytokine induced activity of proteasome subunits** in (**A**) INS-1E, (**B**) mouse and (**C**) human islets (all n = 3). Cells and islets were exposed to low IL-1β dose (30 pg/ml for human islets, 50 pg/ml for mouse islets and 15 pg/ml for INS-1E for 24 hours), high IL-1β dose (300 pg/ml for human islets, 300 pg/ml for mouse islets and150 pg/ml for INS-1E for 24 hours), INF-γ (10 ng/ml for human islets, 10 ng/ml for mouse islets and INS-1E for 24 hours) or control media. Statistical analysis was performed by paired t-tests of treatments versus control. Experiments done on individual islet donors (B and C) or biological cell replicates (A) are connected by lines.

The Human Protein Atlas RNA-seq data indicated constitutive expression of the inducible β1i, β2i and β5i subunits in human pancreas, with 10% of the RNA sequencing reads originating from the islets of Langerhans and 75% coming from exocrine glandular cells [50]. Immunostaining of the islets for specific inducible subunits detected β5i protein by one of two employed antibodies, while β1i and β2i proteins were not detected [50]. To clarify this issue, we first investigated the expression of inducible proteasome subunits in unstimulated human dispersed islet cells. Re-analysis of the previously published data set of single-cell RNA sequencing [37] uncovered substantial subpopulations of α-, β- and δ-cells that constitutively

express all inducible subunits (Fig 1A and 1B) placing human islets on par with other tissues that express those subunits constitutively [9, 16]. When testing β cell models, as well as human and mouse islets, we found that all three inducible β-subunits were detectable at the protein level without the need for cytokine stimulation (Fig 1C–1E), although the expression levels of each subunit varied substantially between tested groups. β cells can therefore potentially assemble i- and int-proteasomes containing one, two or three inducible subunits without immune-stimulation. It is however important to stress that the number of β1i-positive β cells is low in human islets (Fig 1B) thus substantially limiting the possibility to assemble i-proteasome. This observation should be taken into account while investigating proteasome function and composition in pancreatic β cells.

The similar profile of expression of inducible β subunits in β, α and δ cells indicates that those subunits play parallel roles in degradation of the hormones abundantly handled by each cell type ER: insulin, glucagon and somatostatin, respectively. The cellular localization of the i- and int-proteasomes may also play a role in that process, as β5i and β1i subunits are found in close proximity to the ER while the s-proteasomes are homogenously distributed in both nucleus and cytoplasm [51].

Next, we used mass spectrometry to identify the proteasome subtypes in β cells. By immunoprecipitating the α2 subunit from INS-1E cell lysates, we purified active proteasomes. Eighty-six % of total proteasomes contained standard β subunits forming s-proteasomes (Fig 2). The remaining 14% contained only β5i subunit, whereas β1i and β2i could not be detected. Therefore, INS-1E cells constitutively express two types of proteasomes, the s-proteasome and an int-proteasome, where at least one β- ring contains a β5i subunit. We neither detected i-proteasomes nor int-proteasomes with incorporation of inducible subunits other than β5i, despite their expression in unstimulated INS-1E cells. According to the rules of cooperative assembly, β1i cannot be incorporated without β5i but the opposite is feasible [52] and thus it is theoretically possible but biologically less plausible that β1i and β2i are expressed but do not participate in the formation of a pool of active proteasomes. Alternatively, sensitivity of the antibody used for the detection of β1i may be high relative to the sensitivity of the other antibodies used for subunits detection, distorting the evaluation of the intracellular stoichiometry of the inducible subunits. The MS data that failed to detect β1i and β2i incorporation into proteasomes may thus be a more valid measure of the actual subunit stoichiometry in INS-1E cells. Furthermore, according to the rules of proteasome assembly, the lack of incorporation of β1i would prevent the incorporation of β2i subunit [52]. However, we cannot rule out that β1i and β2i are incorporated but constitute the minor portion of active proteasomes, below the detection limit of our MS method.

The enzymatically active proteasomes are generally capped on one or both ends of the central 20S proteasomal core by regulatory particles 19S or 11S, but the method employed in our study does not distinguish which type of proteasome is associated with a given regulatory particle. We have found that 56% of proteasomes in INS-1E cells contained the 19S particle known to associate with all types of proteasomes [21, 46, 53, 54], while 5.1% proteasomes contained 11S that preferentially associates with int- and i-proteasomes [12, 21, 46, 55]. This would indicate that about one third of all INS-1E int-proteasomes are bound to 11S particles while the other two third is associated with 19S particle, (hybrid int-proteasome) or not associated with any regulatory particle and thus presumably not active. Regulatory particles dictate substrate availability and specificity with 19S recognizing client proteins marked by polyubiquitin chains and 11S being involved in the degradation of short and non-ubiquitinated peptides and antigen processing for MHC I presentation [56]. As a result, their presence within assembled and active proteasomes demonstrates that unstimulated β cells contain specialized and mixed populations of proteasomes, possibly reflecting functional specificity.

When profiles of proteasome substrate-specific activities were analyzed, we found that the islets and cell lines (with the exception of A20) all showed similar proteolytic profiles. Chymotrypsin-like activity constituted between 50 and 60% of the total proteasome activity, while trypsin- and caspase-like activities were responsible for the remaining 40–50% (Fig 3A–3C). Immune cell lines have generally been reported to express a higher basal level of inducible subunits, and the i-proteasome constitutes a dominant form of their proteasomes [8]. The fact that INS-1E cells and islets share a similar proteasomal catalytic activity profile indicates that inducible subunits codetermine the activity profiles not only in immune cells.

Off note, we have observed clear differences in the proteasomal catalytic activities in two tested immune cells models, Jurkat and A20. The latter cell profile indicates persistent if not dominant incorporation of β1 subunit (with caspase-like activity) but not β1i (chymotrypsin-like activity) and diminished activity of β5 and/or β5i subunits (chymotrypsin-like activity) that are obligatory part of active proteasomes [56]. The reasons for observed differences are not known but may indicate cancer-cell-specific adjustments, human (Jurkat) vs mouse (A20) divergence or reflect more physiologically important differences between T (Jurkat) and B (A20) cells. Finally, it is plausible that proteasome activity in A20 cells is additionally modified by e.g. post-translational modifications or altered transcription of proteasomal activators, as reviewed in [57].

We next pretreated islets and cells with a β5i selective small-molecule inhibitor, ONX-0194, and found a 30 to 50% reduction in chymotrypsin-like activity, further indicating that the β5i subunit is proteolytically active in β cells (Fig 3D–3F).

Interpretation of the pathophysiological consequences of the proteasomes diversity in β cells requires better understanding of factors influencing its expression and composition. The human genes coding for β5i and β1i map to chromosome 6 precisely between the DNA sequences coding for human leucocyte antigen (HLA)-DQ, HLA-DM and Transporter 1 ATP Binding Cassette Subfamily B Member (TAP) 1 and 2 (S2 Fig), genes known to be major determinants of antigen presentation and predisposing to autoimmune diseases, including type 1 diabetes [58]. The promoter region of β5i contains binding sites for the NFκB transcription factor (S2 Fig), but high concentrations of IL-1β, a strong inducer of NFκB [59], do not increase β5i expression [20]. Accordingly, studies in neurons have shown that high concentration of IL-1β induces Early growth response 1 protein (Egr1) that strongly inhibits transcriptional activity at the β5i promoter [60]. At the same time, Freudenburg et al. speculated that since viral infections induce IL-1β synthesis, iNOS expression and nitric oxide production impeding on mitochondrial function, the resulting reduction in ATP levels would trigger i-proteasome activation and generation of altered peptides that may be immunogenic and enable killing of infected target cells as an appropriate host antiviral response [18]. How can these apparently disparate IL-1β functions be reconciled? We suggest that cytokine concentration and/or duration of exposure are a key determinants of cell fate. It has been reported that low IL-1β concentrations (0.01–0.1 ng/ml) are stimulating and protective for β cells e.g. they improve insulin biosynthesis and secretion and increase β cell proliferation, while higher concentrations (5–20 ng/ml) can induce cell apoptosis and necrosis through e.g. induction of endoplasmic reticulum and mitochondrial stress [47, 48]. These two outcomes employ different cellular pathways, the stimulatory pathway depending on PKC and phospholipases and the toxic pathway on NFκB signaling. We therefore used 10–100 fold lower IL-1β concentrations in our experiments compared to previous publications [20] as well as, in case of mouse islet used for bulk sequencing (Table 3), we extended the islet exposure to IL-1β up to 10 days in order to better mimic long-term low-grade inflammation. Indeed, we found that treatment of human and mouse islets and INS-1E cells with low concentrations of IL-1β increased the mRNA (Table 3) and protein expression of all inducible proteasome β subunits (except of β2i

in human islets, Fig 4), while it had no impact on mRNA levels of standard subunits (Table 3) and increased all substrate-specific proteolytic activities in human and mouse islets and INS-1E cells (Fig 5). Off note, 10 day mouse islets exposure to low concentrations of IL-1β did diminished their glucose induced insulin secretion but did not reduced islets insulin content nor induced endoplasmic reticulum stress or cell death as reported in Ibarra et al. Mol Cell Endocrinol. 2019.

Our results indicate that cytokine concentration is critical when evaluating the regulatory role of cytokines in proteasome expression and activity.

Proteasomes process proteins of both endogenous and exogenous origin and produce peptides that are complexed with MHC I. The shift in composition of proteasomes towards i-proteasomes, changes the peptide repertoire from non-immunogenic to immunogenic [18, 61] and can contribute to the progression towards autoimmune diabetes [18, 20, 62, 63]. Importantly, the observed differences in cytokine action may reflect changing conditions in the islet microenvironment during inflammatory or metabolic stress. IL-1β is a central promoter of low-grade inflammation and protection against certain viral infections, including influenza [64]. One of the possible host protective mechanisms engaged by this cytokine could involve expression and assembly of int- and i-proteasomes that would result in an increased presentation of viral antigens and/or modified self-antigens, thereby enabling T effector-cell dependent eradication of infected cells. Similarly, low grade inflammation and local IL-1β production in the islet microenvironment, could facilitate neoepitope presentation by β cells through preferential incorporation of inducible subunits to form int- or i-proteasomes. Interestingly, cells deficient in β5i show lower MHC I expression and peptide presentation, and β5i pharmacological inhibition slows disease progression in mouse models of inflammatory diseases such as arthritis and lupus [35, 65]. Furthermore, β5i has also been implicated in type 1 diabetes, and its inhibition has been shown to have a protective effect [35]. This could reflect the fact that the MHC I peptides repertoire is at least in part dependent upon the activity of β5i subunit in β cells expressing int- or i-proteasomes.

The role of immune- and, especially, intermediate proteasomes in β cell pathophysiology remains to be uncovered in detail. The perspective that differential proteasome subunit expression dictates the repertoire of β-cell neoepitopes presented by MHC I deserves future investigation. Discoveries in this field could lead to targeted proteasome inhibition as treatment options in diseases with an autoimmune component. In this study, we lay the groundwork for such future investigations. For the first time, we show that int-proteasomes are constitutively expressed and active in β cells and that inducible proteasome subunits can be upregulated in β cells in response to stimulatory low concentrations of IL-1β along with increases in total proteasome activities.

## Supporting information

**S1 Fig. Cytokine induced increase in proinsulin levels in INS-1E cells.** Lysates of cells exposed to IL-1β at low (15 pg/ml) and high concentration (150 pg/ml), IFN-γ (10ng/ml) and control medium, were run on SDS-PAGE and subjected to Western blotting in A) and proinsulin band intensity normalized to tubulin in B). C) and D) Mouse islets and INS1-E cell viability was tested. Staining reagent (AlamarBlue) was added to the cell culture for 4 h, incubated at 37oC, and the resulting fluorescence was read on a plate reader. Statistical analysis was performed by paired t-tests of treatments versus control. The data is shown as means with SD. (TIF)

**S2 Fig. Genetic localization of β5i and β1i genes.** Genes for β5i (PSMB8) and β1i (PSMB9) reside in the MHC-II region on human chromosome 6. Presented transcription factors were included in UCSC genome browser genome GRCh37 and visualized with integrated regulation

from ENCODE-track option. For clarity and in relevance to the current publication, only some transcription factors are presented.
(TIF)

## Acknowledgments

The authors are thankful to JDRF award 31-2008-416 (ECIT Islet for Basic Research program) for providing human islets for this study. The authors also acknowledge C. Wollheim and P. Maechler (University Medical Centre, Geneva, Switzerland) for providing INS-1E cell line. Soren Buus, (Department of International Health, Immunology and Microbiology, Copenhagen, Denmark) generously provided the A20 and Jurkat cells. We would like to thank Nils Billestrup for sharing and discussing his islet sequencing data that is presented in this manuscript.

## Author Contributions

**Conceptualization:** Danielle Verstappen, Dusan Zivkovic, Marie-Pierre Bousquet-Dubouch, Thomas Mandrup-Poulsen, Michal Tomasz Marzec.

**Data curation:** Muhammad Saad Khilji, Danielle Verstappen, Michala Cecilie Burstein Prause, Phillip Alexander Keller Andersen, Dusan Zivkovic, Marie-Pierre Bousquet-Dubouch, Björn Tyrberg, Thomas Mandrup-Poulsen.

**Formal analysis:** Muhammad Saad Khilji, Danielle Verstappen, Tina Dahlby, Michala Cecilie Burstein Prause, Celina Pihl, Sophie Emilie Bresson, Tenna Holgersen Bryde, Phillip Alexander Keller Andersen, Dusan Zivkovic, Marie-Pierre Bousquet-Dubouch, Björn Tyrberg, Thomas Mandrup-Poulsen, Michal Tomasz Marzec.

**Funding acquisition:** Michal Tomasz Marzec.

**Investigation:** Muhammad Saad Khilji, Danielle Verstappen, Tina Dahlby, Michala Cecilie Burstein Prause, Sophie Emilie Bresson, Tenna Holgersen Bryde, Phillip Alexander Keller Andersen, Kristian Klindt, Marie-Pierre Bousquet-Dubouch, Björn Tyrberg, Thomas Mandrup-Poulsen, Michal Tomasz Marzec.

**Methodology:** Muhammad Saad Khilji, Danielle Verstappen, Tina Dahlby, Michala Cecilie Burstein Prause, Celina Pihl, Sophie Emilie Bresson, Tenna Holgersen Bryde, Phillip Alexander Keller Andersen, Kristian Klindt, Dusan Zivkovic, Marie-Pierre Bousquet-Dubouch, Björn Tyrberg, Thomas Mandrup-Poulsen, Michal Tomasz Marzec.

**Project administration:** Michal Tomasz Marzec.

**Supervision:** Marie-Pierre Bousquet-Dubouch, Thomas Mandrup-Poulsen, Michal Tomasz Marzec.

**Validation:** Muhammad Saad Khilji, Sophie Emilie Bresson, Marie-Pierre Bousquet-Dubouch.

**Visualization:** Danielle Verstappen, Celina Pihl, Sophie Emilie Bresson, Tenna Holgersen Bryde, Dusan Zivkovic, Marie-Pierre Bousquet-Dubouch, Björn Tyrberg, Michal Tomasz Marzec.

**Writing – original draft:** Muhammad Saad Khilji, Danielle Verstappen, Tina Dahlby, Michala Cecilie Burstein Prause, Celina Pihl, Sophie Emilie Bresson, Tenna Holgersen Bryde, Kristian Klindt, Dusan Zivkovic, Marie-Pierre Bousquet-Dubouch, Björn Tyrberg, Thomas Mandrup-Poulsen, Michal Tomasz Marzec.

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
