## [Decision Letter · Decision Letter 0]

6 Nov 2019

PONE-D-19-24067

The intermediate proteasome is constitutively expressed in pancreatic beta cells and upregulated by stimulatory, non-toxic concentrations of interleukin 1 b

PLOS ONE

Dear Prof. Marzec,

Thank you for submitting your manuscript to PLOS ONE. After careful consideration, we feel that it has merit but does not fully meet PLOS ONE’s publication criteria as it currently stands. Therefore, we invite you to submit a revised version of the manuscript that addresses the points raised during the review process.

As pointed out by the reviewers, a number of issues could be addressed or at least clarified in the text.

We would appreciate receiving your revised manuscript by Dec 16 2019 11:59PM. To enhance the reproducibility of your results, we recommend that if applicable you deposit your laboratory protocols in protocols.io, where a protocol can be assigned its own identifier (DOI) such that it can be cited independently in the future. For instructions see: http://journals.plos.org/plosone/s/submission-guidelines#loc-laboratory-protocols

We look forward to receiving your revised manuscript.

Kind regards,

Corentin Cras-Méneur, Ph.D.

Academic Editor

PLOS ONE

Journal Requirements:

3. Please amend the manuscript submission data (via Edit Submission) to include author Nils Billestrup.

Reviewers' comments:

Reviewer's Responses to Questions

**Comments to the Author**

1. Is the manuscript technically sound, and do the data support the conclusions?

Reviewer #1: Partly

Reviewer #2: Yes

2. Has the statistical analysis been performed appropriately and rigorously? 

Reviewer #1: Yes

Reviewer #2: Yes

3. Have the authors made all data underlying the findings in their manuscript fully available?

Reviewer #1: Yes

Reviewer #2: Yes

4. Is the manuscript presented in an intelligible fashion and written in standard English?

Reviewer #1: Yes

Reviewer #2: Yes

5. Review Comments to the Author

Reviewer #1: This is a descriptive study that examines the expression of subunits of the immune proteasome in pancreatic islets. The authors have used previously published RNAseq evidence to suggest that components of the immune proteasome are expressed at the mRNA level in b-cells and other islet endocrine cells. Overall the study is consistent with other reports showing immune proteasome expression under various conditions with new information suggestive that long exposures with low concentrations of IL-1 are capable of stimulating subunit expression. There are a number of concerns that could be addressed or clarified in the text. Using RNAseq the authors provide in support of immune proteasome expressed in b-cells under basal conditions. How was RNAseq used to quantify mRNA levels? There is also some concern with the MS analysis of cross-linked proteasomes via immunoprecipitation approaches. It is not clear how the crosslinking on intact cells crosslinks the proteasome found in the cytoplasm (Fig 2). Some controls could be helpful with this analysis to support the authors conclusions.

There is also concern regarding “non-toxic” concentrations of IL-1. It is not clear that mouse or human islets are sensitive to IL-1 alone and therefor all concentrations of IL-1 are non-toxic. It is suggested that the text be modified to low and high concentrations to be more consistent with the biological action. Also- the merits of a 10 day exposure are not clear- what is the time- and concentration-dependence of IL-1 on immune proteasome subunit expression?

Quantification of RNAseq? How is it possible to determine a p-value on these studies (Table 3) and how was quantification performed. Number of cells expressing the targets or the level of expression (i.e. number of reads?). Further, was this confirmed by PCR? Also, was this one isolation of islets and the n represent the number of cells sequenced, or were multiple isolations performed for each single cell sequence?

Reviewer #2: The authors describe the constitutive expression of intermediate proteasomes in pancreatic beta cells. This expression is upregulated by interleukin 1b (IL1b). Mining of available RNA-seq data of single cells from pancreatic islets of healthy individuals and analysis of human and mouse islets, INS-1E (insulinoma), A20 (lymphoma) and Jurkat (leukemic cells) cells, revealed high expression of the inducible proteasome subunits in immune cells and low but consistent expression in the pancreatic cells. By LC-MS/MS analysis, proteasomes from INS-1E cells were classified either as standard (s-proteasomes) or beta5i-containing (14%) but no proteasomes were detected with the beta1i and beta2i subunits. With the exception of the A20 cells, all other cells types tested had proteasomes exhibiting mainly chymotrypsin-like activity (50-60%) followed by trypsin- and caspase-like activity. Incubation of human/mouse islets with ONX-0914 – an inhibitor of beta5i – led to reduction in chymotrypsin-like activity. Moreover, IL-1b upregulated the expression of the inducible subunits beta-1 (b1i), -2 (b2i) and -5 (b5i) and the proteasome activity in beta cells.

The findings are important as our knowledge of the proteasome activities in beta cells is still fairly limited. The authors provide a comprehensive picture of proteasome composition, especially with respect to the inducible subunits, in beta cells. Overall, the study is technically sound and the results support the conclusions. Statistical analysis of the data has been performed. I only have a few minor comments:

1. The authors should address the discrepancy in the level of chymotrypsin-like activity in the proteasomes of beta cells and Jurkat cells vs. A20 cells (lines 305-307). This is only mentioned in passing in the discussion (line 452).

2. Reporting the viability of cells (e.g. in figure 5) would be helpful to better assess the reported effects and the effect of treatment on the cells or islets.

6. PLOS authors have the option to publish the peer review history of their article (what does this mean?). If published, this will include your full peer review and any attached files.

Reviewer #1: No

Reviewer #2: No

---

## [Author Response · Author response to Decision Letter 0]

26 Nov 2019

PONE-D-19-24067

The intermediate proteasome is constitutively expressed in pancreatic beta cells and upregulated by stimulatory, non-toxic concentrations of interleukin 1 b

Response to Reviewers:

Reviewer #1: This is a descriptive study that examines the expression of subunits of the immune proteasome in pancreatic islets. The authors have used previously published RNAseq evidence to suggest that components of the immune proteasome are expressed at the mRNA level in b-cells and other islet endocrine cells. Overall the study is consistent with other reports showing immune proteasome expression under various conditions with new information suggestive that long exposures with low concentrations of IL-1 are capable of stimulating subunit expression.

There are a number of concerns that could be addressed or clarified in the text. Using RNAseq the authors provide in support of immune proteasome expressed in b-cells under basal conditions. How was RNAseq used to quantify mRNA levels?

Two types of RNAseq data are included in the manuscript. Fig.1A presents human single cell sequencing data and Table 3 presents bulk mouse pancreatic islet RNA sequencing data. 

Quantification of single-cell RNA sequencing of human pancreatic islets was performed through data analysis with bcbio-nextgen toolkit (https://bcbio-nextgen.readthedocs.io/en/latest/), followed by the Salmon algorithm (Patro, Duggal et al. 2017) for quantitation of mRNA counts. Data is then expressed as log2 of counts per million (CPM). The original use of data and more analysis details are included in, cited in the Material and Methods section, reference (Segerstolpe, Palasantza et al. 2016). Material and Method section has been amended to reflect changes.

Table 3 and its description (as well as manuscript text at the lane 203-216; all lane references refer to marked copy of the manuscript) has been modified to underlie the fact that data comes from bulk mouse islet sequencing (see below).

Quantification of RNAseq? How is it possible to determine a p-value on these studies (Table 3) and how was quantification performed. Number of cells expressing the targets or the level of expression (i.e. number of reads?). Further, was this confirmed by PCR? Also, was this one isolation of islets and the n represent the number of cells sequenced, or were multiple isolations performed for each single cell sequence?

We thank the Reviewer for raising this issue as it made us realize that the data should be reported in a different and more detailed way. Currently the data reports on in RPKM for the specific genes of 3 independent experiments as well as their means and is analyzed by Student’s paired t-test. The conclusions remain the same. As mentioned in the material and methods section Table 3 reports on whole islets bulk RNAseq and was obtained from 3 separate isolations of islets from several mice per condition. The method of quantification is described in detail and a reference to the original publication is provided. The expression of six genes reported in this study have not been validated by PCR but multiple other genes in the original study were for ex. insulin, Mafa, Pdx-1 and Ucn3.

Text changes can be found in lanes 203-216, 354-355 and overhauled Table 3 at lane 346.

There is also some concern with the MS analysis of cross-linked proteasomes via immunoprecipitation approaches. It is not clear how the crosslinking on intact cells crosslinks the proteasome found in the cytoplasm (Fig 2). Some controls could be helpful with this analysis to support the authors conclusions.

Co-author Marie-Pierre Bousquet’s group has set the original protocol to in vivo crosslink proteins on intact cells of various origins and then to very efficiently immmunopurify proteasome complexes after cell lysis. They have shown that this crosslinking strategy helps stabilizing Proteasome Interacting Proteins (PIPs), in particular important proteasome regulators, like the 19S and PA28 activators, and enables to characterize the whole diversity of proteasome complexes. In their 1st publication (Bousquet-Dubouch MP et al., Mol. Cell. Proteomics, 2009), the comparison of crosslinked cells and non-crosslinked cells (used as a control) has clearly demonstrated the benefit of formaldehyde crosslinking step on the efficacy of recovery of proteasome subunits and PIPs. Since then, the protocol has been successfully applied to study proteasome complexes in different subcellular compartments in leukemic cells (Fabre B. et al, Mol. Cell. Proteomics, 2013) and in cells and tissues of various origins (Fabre B. et al, J. Proteome Res., 2014; Fabre B. et al, Mol. Sys. Biol., 2015).

Material and method section has been amended to include the original publication for the protocol reference (lane 234).

There is also concern regarding “non-toxic” concentrations of IL-1. It is not clear that mouse or human islets are sensitive to IL-1 alone and therefor all concentrations of IL-1 are non-toxic. It is suggested that the text be modified to low and high concentrations to be more consistent with the biological action. 

We agree with the reviewer and have corrected the text accordingly.

Also- the merits of a 10 day exposure are not clear- what is the time- and concentration-dependence of IL-1 on immune proteasome subunit expression?

We thank the Reviewer for raising this issue. The rational for doing the 10 days exposure to low doses of IL-1 was to mimic and better resemble the low-grade inflammation observed in diabetes. Although, even 24h exposure to IL-1 induces significant changes in proteasome subunit expression and activity the longer exposure, more characteristic for diabetes pathology could result in a loss of inducible subunits expression and induction of other detrimental to beta cells processes. However as reported in our collaborators publication (Ibarra et al. Mol Cell Endocrinol. 2019) prolonged exposure of mouse islets to IL-1 resulted in decrease of glucose stimulated insulin secretion but did not reduce cellular insulin content, did not induce endoplasmic reticulum stress or cell death and inducible subunits expression was maintained. This experiment provides a time perspective underlying the relevance of the observed short- and long-term subunits induction to the pathology of diabetes. We have modified the manuscript text at lane 502-511.

Reviewer #2: The authors describe the constitutive expression of intermediate proteasomes in pancreatic beta cells. This expression is upregulated by interleukin 1b (IL1b). Mining of available RNA-seq data of single cells from pancreatic islets of healthy individuals and analysis of human and mouse islets, INS-1E (insulinoma), A20 (lymphoma) and Jurkat (leukemic cells) cells, revealed high expression of the inducible proteasome subunits in immune cells and low but consistent expression in the pancreatic cells. By LC-MS/MS analysis, proteasomes from INS-1E cells were classified either as standard (s-proteasomes) or beta5i-containing (14%) but no proteasomes were detected with the beta1i and beta2i subunits. With the exception of the A20 cells, all other cells types tested had proteasomes exhibiting mainly chymotrypsin-like activity (50-60%) followed by trypsin- and caspase-like activity. Incubation of human/mouse islets with ONX-0914 – an inhibitor of beta5i – led to reduction in chymotrypsin-like activity. Moreover, IL-1b upregulated the expression of the inducible subunits beta-1 (b1i), -2 (b2i) and -5 (b5i) and the proteasome activity in beta cells.

The findings are important as our knowledge of the proteasome activities in beta cells is still fairly limited. The authors provide a comprehensive picture of proteasome composition, especially with respect to the inducible subunits, in beta cells. Overall, the study is technically sound and the results support the conclusions. Statistical analysis of the data has been performed. I only have a few minor comments:

1. The authors should address the discrepancy in the level of chymotrypsin-like activity in the proteasomes of beta cells and Jurkat cells vs. A20 cells (lines 305-307). This is only mentioned in passing in the discussion (line 452).

We have made changes to the manuscript to clearly underlie observed discrepancies and pointed to their possible sources although we have to admit that at this stage we do not understand the observed proteasome profile in A20 cells (lane 466-477). The proteasome activity and/or substrate specific activity can be regulated on multiple levels and the issue has been recently reviewed in Kors et al. Front. Mol. Biosci., 16 July 2019, https://doi.org/10.3389/fmolb.2019.00048, although, again, it does not provide a clear explanation of our observation. 

2. Reporting the viability of cells (e.g. in figure 5) would be helpful to better assess the reported effects and the effect of treatment on the cells or islets.

We agree with the Reviewer and have added data on mouse islets and INS1-E cell viability to the S1 figure and made appropriate changes in the manuscript text (lane 367-368) adding also references from previously published work e.g. Ibarra et al. Mol Cell Endocrinol. 2019 PMID: 31362031 Fig. 1E. 

References

Patro, R., G. Duggal, M. I. Love, R. A. Irizarry and C. Kingsford (2017). "Salmon provides fast and bias-aware quantification of transcript expression." Nat Methods 14(4): 417-419.

Segerstolpe, A., A. Palasantza, P. Eliasson, E. M. Andersson, A. C. Andreasson, X. Sun, S. Picelli, A. Sabirsh, M. Clausen, M. K. Bjursell, D. M. Smith, M. Kasper, C. Ammala and R. Sandberg (2016). "Single-Cell Transcriptome Profiling of Human Pancreatic Islets in Health and Type 2 Diabetes." Cell Metab 24(4): 593-607.

---

## [Editor Report · Decision Letter 1]

2 Jan 2020

PONE-D-19-24067R1

The intermediate proteasome is constitutively expressed in pancreatic beta cells and upregulated by stimulatory, low concentrations of interleukin 1 b

PLOS ONE

Dear Prof. Marzec,

Thank you for submitting your manuscript to PLOS ONE. After careful consideration, we feel that it has merit but does not fully meet PLOS ONE’s publication criteria as it currently stands. Therefore, we invite you to submit a revised version of the manuscript that addresses the points raised during the review process.

While most of the comments in the manuscript have been addressed according to the comments made by the reviewers, two points remain to be clarified in the text for the manuscript to be accepted.

We would appreciate receiving your revised manuscript by Feb 16 2020 11:59PM. To enhance the reproducibility of your results, we recommend that if applicable you deposit your laboratory protocols in protocols.io, where a protocol can be assigned its own identifier (DOI) such that it can be cited independently in the future. For instructions see: http://journals.plos.org/plosone/s/submission-guidelines#loc-laboratory-protocols

We look forward to receiving your revised manuscript.

Kind regards,

Corentin Cras-Méneur, Ph.D.

Academic Editor

PLOS ONE

Additional Editor Comments (if provided):

The revised version of the manuscript addresses most of the comments raised by the original reviewers.

There are only two points that remained to be clarified in the manuscript:

• The authors need to mention and justify that Normal distribution needs too be assumed for the use of parametric tests.

• The authors need to emphasize and discuss in the text that the replicates are technical replicates (3 isolations) and not biological replicates.

---

## [Author Response · Author response to Decision Letter 1]

21 Jan 2020

PONE-D-19-24067

The intermediate proteasome is constitutively expressed in pancreatic beta cells and upregulated by stimulatory, non-toxic concentrations of interleukin 1 b

Response to Editor Comments 

The revised version of the manuscript addresses most of the comments raised by the original reviewers.

There are only two points that remained to be clarified in the manuscript:

• The authors need to mention and justify that Normal distribution needs too be assumed for the use of parametric tests.

The following text has been added to Statistical analysis section of the manuscript:

All samples were selected without bias and represent biological not technical variations. Distribution of islets, specifically, were randomized and independent of e.g. size and shape. As a result, samples should be homogenous and represent biological variation, and both protein expression and activity is therefore assumed to be normally distributed (43, 44). Furthermore, normality of all expression data was tested with a Shapiro-Wilk test and found normally distributed and tested using a student t-test. Meanwhile proteasome activity and cell viability data is represented by a mean value of technical replicates, and as such should be normally distributed according to the central limits theorem (45).

• The authors need to emphasize and discuss in the text that the replicates are technical replicates (3 isolations) and not biological replicates.

The replicates presented in all figures, irrespective if those are cells or pancreatic islets, are biological not technical replicates. 

Islet isolation: islets were isolated on separate days and from separate group of mice and that constitutes biological replications. Similarly, in case of human islets obtained from deceased donors, isolated islets represent biological variability (thus are replications) not a technical replicate. 

Proteasome studies: for each data point cells were separately cultured (minimum two weeks), plated (on different days) and collected. That procedure constitutes biological replication for cell culture experiments. Then each data set was run, in minimum, as three technical replicates and the mean of those is presented. 

We have amended the manuscript text to emphasize the fact that presented data is based on biological replicates. See lanes: 164-164, 287, 311 (already mentioned), 334-335, 361-362, 383, 403 (already mentioned).

---

## [Editor Report · Decision Letter 2]

22 Jan 2020

The intermediate proteasome is constitutively expressed in pancreatic beta cells and upregulated by stimulatory, low concentrations of interleukin 1 b

PONE-D-19-24067R2

Dear Dr. Marzec,

We are pleased to inform you that your manuscript has been judged scientifically suitable for publication and will be formally accepted for publication once it complies with all outstanding technical requirements.

With kind regards,

Corentin Cras-Méneur, Ph.D.

Academic Editor

PLOS ONE
---

## [Editor Report · Acceptance letter]

5 Feb 2020

PONE-D-19-24067R2 

The intermediate proteasome is constitutively expressed in pancreatic beta cells and upregulated by stimulatory, low concentrations of interleukin 1 β 

Dear Dr. Marzec:

I am pleased to inform you that your manuscript has been deemed suitable for publication in PLOS ONE. Congratulations! Your manuscript is now with our production department. 

With kind regards,

on behalf of

Dr. Corentin Cras-Méneur 

Academic Editor

PLOS ONE